# The Influence of Physical Fields (Magnetic and Electric) and LASER Exposure on the Composition and Bioactivity of Cinnamon Bark, Patchouli, and Geranium Essential Oils

**DOI:** 10.3390/plants13141992

**Published:** 2024-07-21

**Authors:** Camelia Scheau, Carmen Rodica Pop, Ancuța Mihaela Rotar, Sonia Socaci, Anamaria Mălinaș, Marius Zăhan, Ștefania Dana Coldea, Viorel Cornel Pop, Nicodim Iosif Fit, Flore Chirilă, Horia Radu Criveanu, Ion Oltean

**Affiliations:** 1PhD School of Agricultural Engineering Sciences, USAMV Cluj-Napoca, 3-5, Mănăştur Street, 400372 Cluj-Napoca, Romania; camelia.scheau@usamvcluj.ro; 2Department of Food Science, Faculty of Food Science and Technology, University of Agricultural Sciences and Veterinary Medicine of Cluj-Napoca, 64, Floresti Street, 400509 Cluj-Napoca, Romania; carmen-rodica.pop@usamvcluj.ro (C.R.P.); anca.rotar@usamvcluj.ro (A.M.R.); sonia.socaci@usamvcluj.ro (S.S.); 3Department of Environmental Protection and Engineering, Faculty of Agriculture, University of Agricultural Sciences and Veterinary Medicine, 3-5, Mănăştur Street, 400372 Cluj-Napoca, Romania; 4Faculty of Animal Science and Biotechnology, University of Agricultural Sciences and Veterinary Medicine of Cluj-Napoca, 3-5, Mănăştur Street, 400372 Cluj-Napoca, Romania; mzahan@usamvcluj.ro (M.Z.); stefania.coldea@usamvcluj.ro (Ș.D.C.); 5Faculty of Veterinary Medicine, University of Agricultural Sciences and Veterinary Medicine, 3-5, Mănăştur Street, 400372 Cluj-Napoca, Romania; 6Department of Paraclinical Sciences, Faculty of Veterinary Medicine, University of Agricultural Sciences and Veterinary Medicine Cluj-Napoca, 3-5, Mănăstur Street, 400372 Cluj-Napoca, Romania; nfit@usamvcluj.ro (N.I.F.); flore.chirila@usamvcluj.ro (F.C.); 7Faculty of Horticulture, University of Agricultural Sciences and Veterinary Medicine Cluj-Napoca, 3-5, Mănăstur Street, 400372 Cluj-Napoca, Romania; criveanuhoria@yahoo.ro; 8Department of Plant Protection, University of Agricultural Sciences and Veterinary Medicine of Cluj-Napoca, 3-5, Mănăstur Street, 400372 Cluj-Napoca, Romania; ion.oltean@yahoo.com

**Keywords:** essential oil, *Cinnamomum zeylanicum* Blume, *Pelargonium graveolens* L’Hér, *Pogostemon cablin* (Blanco) Benth., magnetic field, electric field, laser, antibacterial, antioxidants

## Abstract

In recent years, essential oils (EOs) have received increased attention from the research community, and the EOs of cinnamon, patchouli, and geranium have become highly recognized for their antibacterial, antifungal, antiviral, and antioxidant effects. Due to these properties, they have become valuable and promising candidates for addressing the worldwide threat of antimicrobial resistance and other diseases. Simultaneously, studies have revealed promising new results regarding the effects of physical fields (magnetic and electric) and LASER (MEL) exposure on seed germination, plant growth, biomass accumulation, and the yield and composition of EOs. In this frame, the present study aims to investigate the influence of MEL treatments on cinnamon, patchouli, and geranium EOs, by specifically examining their composition, antimicrobial properties, and antioxidant activities. Results showed that the magnetic influence has improved the potency of patchouli EO against *L. monocytogenes*, *S. enteritidis*, and *P. aeruginosa*, while the antimicrobial activity of cinnamon EO against *L. monocytogenes* was enhanced by the electric and laser treatments. All exposures have increased the antifungal effect of geranium EO against *C. albicans*. The antioxidant activity was not modified by any of the treatments. These findings could potentially pave the way for a deeper understanding of the efficiency, the mechanisms of action, and the utilization of EOs, offering new insights for further exploration and application.

## 1. Introduction

Plants have been around us since our beginning, offering us food, shelter, clothing, medicines, etc. A special category is represented by aromatic plants that produce EOs. These are complex mixtures of different aromatic and volatile substances with specific roles in plant defense, internal regulation, modulation of the plant microbiome, communication, and attraction of pollinators [1,2,3,4]. EOs have been used for their healing properties since ancient times, and numerous studies continue to elucidate their diverse and intricate effects on human health, through their antiseptic, antimicrobial, antifungal [5], antiviral, cytotoxic [6], antiparasitic, analgesic, spasmolytic [7], sedative, anxiolytic, antidepressive, immunomodulatory [8], anti-inflammatory, antioxidant, and anticancer effects [9]. Due to their potent antibacterial activity, certain EOs (such as cinnamon, thyme, clove, rosemary, oregano, lemon, and tea tree) emerge as formidable candidates for combating the escalating prevalence of pathogenic bacteria and the growing threat of antimicrobial resistance, a pressing global concern [10,11,12,13]. Data from the World Health Organization reveals that in 2019, 1.27 million deaths were attributed to antimicrobial resistance, with an additional 4.95 million deaths related to it. The misuse and overutilization of antibiotics in the treatment of humans, animals, and plants has significantly contributed to the emergence of drug-resistant infections [14]. Additionally, due to their numerous side effects and their well-known capacity to disrupt the intestinal microbiome, natural remedies, more in tune with our organisms, are being investigated [15,16,17,18,19]. Moreover, several studies have shown that EOs have a beneficial effect on microbiota while affecting pathogenic bacteria [20,21,22].

Still, in order to produce high-quality EOs, vast areas of land are needed, as large masses of plant materials are required for the extraction of small quantities of EOs, and therefore new methods for improving the cultivation of plants are being investigated [5,23]. Several studies pointed out that the exposure of seeds, seedlings, or plants to magnetic [24,25] and electric fields, [26,27,28] and LASER irradiation [29,30] has produced interesting and promising results regarding seed germination, plant development, and biomass accumulation.

Furthermore, in some cases, these treatments have improved the mineral uptake, the content of vitamins and secondary metabolites, have modified the overall composition of the EOs, and even increased their production yields [31,32,33,34,35,36,37,38,39,40,41]. For example, the exposure of *Ocimmum basilicum* plants to a static magnetic field (SMF) of 30 mT for 6 days at 5 h/day resulted in an increase in the percentage of the extracted EO and in the main compound methyl chavicol, while the percentage of nerol and geraniol decreased [31]. However, the pretreatment of the harvested parsley plants with a SMF for 8 h did not significantly influence the EO yield, but it increased substantially the concentration of myristicin, the main compound, by 28.8% [42]. Hence, these types of treatments could offer a viable solution to optimize the content of active substances in cultivated medicinal plants [31,42].

When considering the impact of the electric field, a study from 2021 on two varieties of kale (*Brassica oleracea var. acephala*) has shown that, compared to the control, plants growing under an intensity of 25.6 V/m exhibited a higher antioxidant capacity (70%), an increase in the total phenolic compounds (57%), and more calcium (72%) [43]. Additionally, laser light treatment (632 nm, 5 mW, 5 min) had a stimulating effect on anise (*Pimpinella anisum*) and lemongrass (*Cymbopogon proximus*) sprouts, leading to an enhancement of the antioxidant, hypocholesterolemic, and antidiabetic activities of the extracted EOs [39,40].

Nevertheless, sometimes results could be confusing and contradictory due to the fact that there is a wide range of testing conditions and great variability in the way plants may react under the same treatments. Additionally, many of the underlying mechanisms for these influences are not yet fully understood [44,45,46].

As there are more and more studies that investigate the effects of such pretreatments on the evolution and development of seeds and plants, according to our knowledge, there is little research available about the way MEL affects the properties of EOs when applied directly to the extracts themselves. Therefore, our research aims to investigate the impact of MEL on the composition and the antibacterial, antifungal, and antioxidant activity of three EOs of cinnamon bark (*Cinnamomum zeylanicum* Blume), patchouli (*Pogostemon cablin* (Blanco) Benth.), and geranium (*Pelargonium graveolens* L’Hér), well known for their antibacterial and antifungal effects.

Cinnamon EO is extracted from the bark of *Cinnamomum zeylanicum* Blume, also known as *Cinnamomum verum* J. Presl, a tree from the *Lauraceae* family that grows in Sri Lanka and the Malabar Coast of India. Cinnamon is a widespread spice that has antimicrobial, anti-inflammatory, antifungal, antiparasitic, antiproliferative, and antiviral activity [47,48,49].

The EO of patchouli is obtained from the upper parts of the plant *Pogostemon cablin* (Blanco) Benth., which belongs to the *Lamiaceae* family and grows in Southeast Asia, Indonesia, and the Philippines. It is used in perfumery and aromatherapy for its relaxant, antitumoral, antimicrobial, anti-inflammatory, antiviral, and anti-emetic effects [50,51,52].

Geranium (*Pelargonium graveolens* L’Hér) is part of the *Geraniaceae* family and has been cultivated for its precious EO that is well used in cosmetics, perfumery, and for treating skin conditions due to its antioxidant, antimicrobial, repellent, anticancer, sedative, and anti-diabetic properties [53,54,55].

Improved bioactivity could lead to a decrease in the dose used, a better utilization of EOs that would lead to an increase in acceptance, tolerance, and compliance in patients, help develop new treatments, and also open new doors to a better understanding of the complex mechanism of action of EOs.

## 2. Results

All samples were investigated through GC-MS and the compounds were identified based on the mass spectra libraries of the NIST27 and NIST147 software. The results are presented below. Additionaly, more details from this analysis as well as the chromatograms for all the EOs were included in the Appendix A: Chromatographic Data for Cinnamon, Patchouli, and Geranium Essential Oils, and Chromatograms of the Essential Oils.

### 2.1. Chemical Composition of the Essential Oils

#### 2.1.1. Cinnamon Bark (*Cinnamomum zeylanicum* Blume) Essential Oils

For the cinnamon bark essential oil (CEO) 26 compounds were identified. Cinnamaldehide, a phenylpropanoid substance, was found to be the main compound, with a percentage of 67.39%. Eugenol, cinnamyl acetate, caryophyllene (sesquiterpene), and β-linalol (monoterpene) were the following substances according to the main percentages. Other substances like α-pinene and α-phellandrene (monoterpenes), α-caryophyllene (sesquiterpene), and eugenol acetate (phenylpropanoid) were also present (Table 1).

Considering the energized EOs from this group, generally, the main profile was kept, with small variations in regard to the percentages or to the very existence of certain compounds. For instance, in all samples of treated EOs, δ-2-Carene was not detected, γ-Terpinene was not present in the cinnamon bark EOs exposed to magnetic and electric fields (CEOM and CEOEl), and α-terpinene and butyric acid were lacking in the cinnamon bark essential oil exposed to laser irradiation (CEOL). Additionally, terpinolene and β-*cis*-Ocimene were not present in the initial sample of CEO but appeared in all samples exposed to MEL.

#### 2.1.2. The Patchouli (*Pogostemon cablin* (Blanco) Benth.) Essential Oils

In the patchouli essential oil (PEO), 17 compounds were identified. Patchoulol was found in the largest quantity (37.70%), followed by α-bulnesene and α-guaiene, which were 19.96% and 15.40%, respectively. Other substances identified included seychellene, α-patchoulene, and caryophyllene, all part of the sesquiterpenes group. Very small traces of α-pinene and β-pinene from the monoterpenes class were also present (Table 2).

In the case of these EOs, all tested samples had very similar chromatographic profiles.

#### 2.1.3. The Geranium (*Pelargonium graveolens* L’Hér) Essential Oils

For geranium essential oil (GEO), 24 substances were identified, including (R)-citronellol, *cis*-geraniol, and citronellol acetate accounting for the main compounds at 44.28%, 20.92%, and 10.16%, respectively. The following molecules were also present in significant quantities: *cis*-menthone (5.98%), β-linalool (4.61%), and *cis*-geranyl acetate (3.79%). While the monoterpenes represent the main compound class of this EO, sesquiterpenes components like caryophyllene (1.87%), β-bourbonene (1.55%), and copaene (1.27%) were also present (Table 3).

When comparing the various samples within this group, the results indicate similar profiles for the main compounds. However, minor differences, primarily in the presence and percentages of minor compounds, are evident.

For example, α-pinene is missing in the geranium essential oil exposed to laser irradiation (GEOL), while *trans*-menthone (1.96%) and (R)-(+)-citronellal, isopregol were not present in the initial sample but were present in all other treated EOs.

### 2.2. Antimicrobial Activity of the Essential Oils

All EOs in all exposures showed antibacterial activity against both Gram-positive and Gram-negative pathogenic strains of bacteria. Furthermore, they have all demonstrated antifungal activity against Candida albicans. Yet, the intensity of these effects varies both between the selected EOs and between the applied treatments. Minimum inhibitory concentration (MIC) and minimum bactericidal concentration (MBC) were measured for each sample and bacterial or yeast strain.

#### 2.2.1. Antimicrobial Activity of the Cinnamon Bark (*Cinnamomum zeylanicum* Blume) Essential Oils

CEO had a powerful antibacterial and antifungal activity against all strains. On top of this, exposing the oil to MEL significantly improved its antibacterial activity against certain pathogenic strains. In the case of *Staphylococcus aureus*, the exposure to magnetic and electric fields reduced MIC and MBC by 50%. For *Listeria monocytogenes,* the best results were obtained for CEOL when MIC had decreased three times and MBC five times, when compared to the untreated CEO. When it comes to *Salmonella enteritidis,* the MIC was slightly improved by exposure to the electric field (CEOEl), while MBC dropped by 50%, which is statistically significant only for MBC. Interesting results were obtained for the strains of *C. albicans*, where CEOM had similar results to CEO, yet the exposure to the electric field has significantly inhibited the antifungal effect of CEOEl (Table 4).

#### 2.2.2. Antimicrobial Activity of Patchouli (*Pogostemon cablin* (Blanco) Benth.) Essential Oils

PEO had an enhanced antimicrobial activity against all strains, except for *S. enteritidis* and *P. aeruginosa,* against which its effect was less intense. When comparing the results for *S. aureus*, the patchouli essential oil exposed to the electric field (PEOEl) had MIC values three times smaller than PEO, while MBC was reduced by more than four times. The exposure to the magnetic field (PEOM) decreased investigated values by half for the same bacterial strain. In the case of *L. monocytogenes,* all energized samples had improved MBC, the best results being recorded for PEOM. While for *S. enteritidis*, all treated EOs had decreased MBC by 50% and the MIC improved only for PEOM and PEOEl, for *P. aeruginosa* the inhibitory capacity was enhanced by all exposures and MBC was diminished only by PEOM and PEOL (patchouli essential oil exposed to laser irradiation). Strong inhibitory effects of the antibacterial activity were observed for PEOEl against the *E. coli* culture. Moreover, the antifungal activity was reduced by all treatments and was more pronounced for PEOEl and PEOL (Table 5).

#### 2.2.3. Antimicrobial Activity of Geranium (*Pelargonium graveolens* L’Hér) Essential Oils

All strains were sensitive to the antimicrobial action of GEO. In the case of *S. aureus,* all samples exposed to MEL have improved both the inhibitory and bactericidal activity of the geranium EO, with the best results obtained by GEOM (MIC was reduced by nine times and MBC by four). The influence of laser irradiation has significantly improved MIC and MBC when the oil was applied to *B. cereus* and *L. monocytogenes* cultures, while it only reduced MIC for *P. aeruginosa*. For the *E. coli* strain, both GEOEl and GEOL have substantially enhanced the antibacterial effect. When it comes to *C. albicans*, results show that all treatments had an enhancement effect on the antifungal activity of the geranium EO. Interestingly, all exposures slightly decreased the antibacterial effect against the *S. enteritidis* strain (Table 6).

### 2.3. Antioxidant Property of the Essential Oils

All investigated EOs in all treatments had great antioxidant activity with similar values when measured through the DPPH and ABTS^+^ tests (Table 7, Table 8 and Table 9). All samples demonstrated a slightly increased antioxidant activity on ABTS^+^.

## 3. Discussion

Essential oils are receiving increased attention from both the general public and the scientific community due to a growing interest in finding natural or plant-based products and solutions for our current health issues [56]. The global EOs market reached USD 23.74 billion in 2023, with forecasts indicating continued growth in the coming years [57]. Owing to their wide range of bioactivities, good tolerance, efficiency, and diversity of applications, EOs could provide valuable support and solutions for the prevention and treatment of many diseases [58]. Additionally, due to their safe profile, they are being researched to be included as additives in the food and cosmetic industries [59].

There is promising evidence that bacterial resistance, which is constantly increasing because of the overuse of antibiotics and is thus considered a current poignant health threat, could be diminished by the use of EOs [12]. Due to their complex composition, it is more difficult for bacteria to develop resistance to multiple molecules that act through various mechanisms, thus exerting a synergistic effect [60]. Additionally, it was observed a synergic action when more EOs were used or when EOs were combined with other medicines like antibiotics [1,61]. Moreover, several studies on animals have shown that the use of EOs like cinnamon, patchouli, oregano, and thyme favors the development of beneficial bacteria in the gut while inhibiting potential pathogenic strains [62,63,64,65,66]. Gut dysbiosis has also been correlated with the incidence of other conditions like cardiovascular diseases, diabetes mellitus, or overweight, and therefore, the use of EOs that provide additional effects (anti-inflammatory, immunomodulating, dyslipidemic, antidiabetic, and antioxidant) could improve overall health [67,68,69,70].

Conjointly, recent research has investigated the influence of MEL, with positive results when it comes to germination, plant development, EO extraction yields, and even the enhancement of certain compounds in the final extract [24,25,26,27,28,29,30,31,32,33,34,35]. One study has investigated the antimicrobial effects of the EOs extracted from plants grown from geranium *(Pelargonium graveolens)* seeds exposed to laser light treatment, which proved to have increased activity against *Bacillus subtilis*, *Streptococcus salivarius*, and *Sarcina lutea* cultures [71]. Another study from 2022 reported that the antibacterial effect of pine (*Pinus sylvestris* L.) EO was amplified when a rotating magnetic field was applied to a saline solution containing a suspension of *E. coli* and EO, showing promising perspectives in this direction [72].

This study aimed to investigate whether applying MEL directly to EOs could influence their composition and bioactivity. As research in this area is currently limited, this could suggest a novel avenue for further investigation, providing new opportunities for exploration and enhancing our understanding of the mechanisms of action of EOs.

According to our findings, MEL exposures exert varying degrees of influence on EOs. Generally, the main chemical profiles of all tested essential oils were pretty similar for each group. However, results showed small variations in percentages, especially for the substances that were present in minor quantities. In some cases, new substances were detected, while others that were present in the pure EO were missing in some energized samples.

For cinnamon and geranium EOs, their composition was altered slightly by these treatments, while the profiles of patchouli EOs were more stable. The gas chromatogram of the initial CEO is aligned with other results mentioned in the literature, which revealed cinnamaldehyde, eugenol, cinnamyl acetate, caryophyllene, and β-Linalool as the main compounds [73,74,75]. However, other studies present different results [76,77], which is another indication that the composition of the EO extracted from the same plant can vary significantly due to the examined species, genotype, chemotype, climate, soil, or extraction method [78,79,80]. The investigated CEO had a composition of 81.27% phenylpropanoids, 12.63% monoterpenes, and 5.95% sesquiterpenes.

CEO has shown powerful antibacterial and antifungal activity against all investigated bacterial strains and *C. albicans*. This could be due to its high content of cinnamaldehyde, a negatively charged phenylpropanoid compound that has the potential to disrupt the biological processes of microorganisms, especially at the level of proteins and nucleic acids that are high in nitrogen [81,82]. Additionally, the presence of eugenol enhances the antimicrobial effect [83]. Research has revealed that CEO suppresses bacteria through anti-quorum sensing activity, by blocking the ATPase, cell division, biofilm development, membrane porine and motility [81]. All these are changing the lipidic profile and damaging the cell membrane, causing auto-aggregation [84,85].

Nevertheless, its intense antimicrobial effect is also attributed to the synergistic effect of more compounds that act on different cell targets, as is the case for most EOs [60,83,85].

Of all the investigated oils, CEO had the best bactericidal effect against *P. aeruginosa*, known for its resistance to many substances. This result is consistent with those of Coșeriu et al., who found that CEO was active against all 72 clinical isolates of *P. aeruginosa* [76]. Other studies support these findings as well as its strong antimicrobial activity [85,86,87,88,89].

In PEO, 97.36% of the molecules are sesquiterpenes with patchoulol being the main compound, followed by α-bulnesene, α-guaiene, seychellene, caryophyllene, and patchoulene, which is in accordance with several studies [90,91,92,93,94]. Still, it was observed that the composition of the examined PEO is variable due to the geographical location, the part of the plant used and the applied extraction method [91,92,95].

In our study, PEO demonstrated great antibacterial and bactericidal effects against *S. aureus*, *B. cereus*, *L. monocytogenes*, and *E. coli*, and it was less active against *P. aeruginosa* and *S. enteritidis*. Additionally, it proved to have a powerful antifungal activity against *C. albicans.* Strong antibacterial and antifungal properties of PEO are consistent with those reported in the literature [83,95,96]. The main antimicrobial activity was attributed mainly to the existence of patchoulol, α-patchoulene, and pogostone, however the latter was not detected in our sample through the utilized method [97].

The investigated GEO has a monoterpenic profile (95.07%) combined with a small percentage of sesquiterpenes (4.69%). The main identified compounds were (R)-citronellol, followed by *cis-*geraniol, which is also in accordance with other findings [98,99,100,101]. The third main detected substance was citronellol acetate, which differs from other studies that mention neryl acetate [98], citronellyl formate [99,102] or isomenthone [53]. GEO had a strong antibacterial effect both on Gram-positive and Gram-negative strains, which were also reported in the literature [101,103,104]. Powerful antifungal activity was also noted for GEO [101,102,103,104] and is attributed to the main compounds through the inhibition of germ tube induction and the destruction of membrane integrity, thus suppressing the cell cycle of *C. albicans* [103].

When analyzing the influence of MEL on antibacterial and antifungal activity, we can observe that the effects vary between the oils and among the specific bacterial or yeast strains they are acting on.

Thus, we have observed that the antibacterial effect against *S. aureus* was enhanced for all investigated oils through exposure to both the magnetic and electric fields, while the laser irradiation has improved the potency only for the geranium EO (GEOL).

Additionally, the exposure to the magnetic field had a favorable effect on patchouli EO when applied to *L. monocytogenes*, *S. enteritidis, and P. aeruginosa* and on geranium EO when acting against *C. albicans*.

The electric field treatment improved the antibiotic activity of cinnamon EO against *L. monocytogenes* and *S. eteriditis* (CEOEl), as well as patchouli EO against *S. eteriditis* and *P. aeruginosa* (only MIC was decreased by PEOEl), and geranium EO against *E. coli* (GEOEl). The antifungal effect was also significantly increased for GEOEl. However, it is interesting to note that the antifungal activity of CEOEl and patchouli PEOEl were notably reduced by the electric field. A strong diminishing influence was also observed for the antibacterial effect of PEOEl against *E. coli*.

The laser irradiation had a stimulating effect on the antibacterial activity for all oils when applied on *L. monocytogenes* and it strongly enhanced GEOL against all strains, except for *S. eteriditis*. It also activated the antibacterial effect of PEOL on *P. aeruginosa*, while its antifungal activity was reduced. Moreover, the magnetic field increased the antibacterial activity of PEOM against *L. monocytogenes* and decreased this effect for GEOM.

A study from 2022 by Kiełbasa et al. has revealed that exposure to an electromagnetic field influences the antiseptic properties of two EOs: tea tree and cedarwood. There was variability in these properties according to the characteristics of the electromagnetic field, the exposure time, the oils used, and the investigated bacterial strain. It was observed that an electromagnetic field of 80 mT had simulative effects on both oils, while a 40 mT field significantly reduced the antimicrobial action of the tea tree EO [105]. Further investigations on cedarwood EO have suggested that the electromagnetic field strength and exposure duration have impacted the spectral properties of cedarwood EO [106].

In order to assess the antioxidant activity of EOs, two common and easy methods were employed: DPPH and ABTS^+^ [107,108]. DPPH is a stable nitrogen-centered free radical that can receive an electron or a hydrogen radical and form a lasting diamagnetic molecule [109]. ABTS^+^ is another nitrogen-centered radical with a distinctive blue-green hue, which is reduced to its nonradical, colorless (ABTS) form by substances with antioxidant activity [108].

When comparing the antioxidant activity of the investigated samples, we could notice that there was no difference between the applied treatments. Moreover, there was no recorded difference between the tested oils, even though they were extracted from plants that belong to different families and have a different chemical profile: phenylpropanoidic in the case of CEO, sesquiterpenic for PEO, and monoterpenic for GEO. We could also notice that all oils, regardless of the applied treatment, had slightly enhanced antioxidant activity on ABTS^+^ than on DPPH.

In several studies, CEO proved to have powerful antioxidant activity, which was correlated with the high dose of eugenol, a phenolic substance, more likely than cinnamaldehyde [80,109,110]. Various IC50s for DPPH scavenging activity by cinnamon EOs were reported: 1.1 µg/mL [80], 0.03 µg/mL [87], 0.41 mg·mL^–1^ [109], and 0.30 ± 0.06 mg/mL [111]. In a study by Banglao, W., CEO showed strong antioxidant activity that was directly proportionate with the concentration used, both for DPPH and ABTS^+^. Moreover, CEO seemed to present a better antioxidant capacity on ABTS^+^ than on DPPH, with an inhibition of 20.10 ± 1.71% for DPPH and 22.01 ± 0.90% for ABTS^+^ at a concentration of 500 µg/mL, which confirmed our findings [112].

PEO is also presented in the literature with excellent antioxidant activity, despite the inconsistencies related to the reported data, which makes the comparison difficult [52,108]. For instance, for DPPH, Mansuri et al. has found an IC50 of 19.53 mg/mL [94], while Soh et al. mentioned IC50 of different samples of essential oils to be between 0.42 and 1.92 mg/mL [113] and Hariyanti et al. presented an IC_50_ of 22.45 μg/mL for PEO from the region of Cibinong and of 19.87 μg/mL from Batu [114]. Galovičová et al. suggests an antioxidant activity of 71.4 ± 0.9% [52]. Paulus et al. established the free-radical scavenging activity of PEO at 12.08 µmol Trolox/mL [115]. The antioxidant activity of PEO is mainly attributed to high concentrations of patchoulol [51,114].

In a wider study on 42 EOs, the antioxidant activity of DPPH at a concentration of 5 mg/mL of CEO was found to be 91.4 ± 0.002%, while for PEO it was 15.63 ± 0.009%, which is contradictory to our results in regard to PEO [110]. The same study found a correlation between the antioxidant activity and the phenolic composition. However, Pandey et al. reported powerful antioxidant activity for PEO, both for the EO extracted from leaves (99.43%) and for the one extracted from flowers (91.04%). These values were recorded at a concentration of 30 µg/mL [51]. Dechayont et al. investigated the scavenging potential of *P. cablin* extracts through DPPH and ABTS^+^ assays and found that the ethanolic extract had the strongest antioxidant activity (IC50_DPPH_ = 18 ± 0.90 and IC50_ABTS =_ 20 ± 0.24 µg/mL), with a slightly improved effect on DPPH [97].

GEO is also reported to have good antioxidant activity, mainly due to the presence of oxygenated monoterpenes like the principal compounds citronellol and geraniol [116,117,118]. Regarding the free-radical scavenging activity of GEO on DPPH, studies have shown an IC50 of 0.802 mg/mL as reported by Džamić et al. [102] and 18.02 μg/mL according to Lohani et al. [117]. Another study by Ćavar has revealed that the GEO from leaves had a slightly better free radical scavenging activity than the one extracted from stems, with an IC50 of 63.70 ± 1.56 mg/mL and 64.88 ± 1.12 mg/mL, respectively [118]. However, in this case, the results show a diminished antioxidant activity when compared with our findings. The study of Dumlupinar et al. highlighted that the capacity to scavenge the DPPH radical is directly proportionate with the concentration, as follows: at a concentration of 10 µg/mL, the inhibition was the smallest (20.54%), while at a concentration of 200 µg/mL, it was 82.4% [53].

When comparing the antioxidant activity of DPPH with the one on ABTS^+^, Kačániová et al. report that GEO proved a more potent antioxidant activity in the ABTS^+^ test with an IC50 value of 0.26 ± 0.02 mg/mL, while in the DPPH test the IC50 was measured at 1.14 ± 0.08 mg/mL [116]. The results from our study showed a similar antioxidant effect on both radicals, with a slightly higher activity on ABTS^+^ (2.18 ± 0.005 µM TE/mL) than on DPPH (1.85 ± 0.74 µM TE/mL).

While there are more and more studies investigating the antimicrobial and antioxidant effects of EOs and their mechanisms of action, the direct influence of MEL on the EOs is just beginning to be researched. Our study, along with the one by Kiełbasa [105], shows that various MEL affect differently the antibacterial, bactericidal, and antifungal properties of EOs, some of them enhancing statistically significant changes in their activity. Yet, our understanding of the mechanisms behind these effects is still limited. We could notice that, when exposed to MEL, the cinnamon EO and geranium EO were more susceptible to modest changes in the composition, which were more visible at the level of the minor compounds and at very small percentages. Under the same treatment, the patchouli EO showed a more stable profile.

As the changes in composition were negligible, the observed effects induced by these treatments might be caused by dynamics that we are still unaware of, which deserve to be investigated in future studies. Furthermore, the influence of MEL varies according to the type of exposed EOs, the microbial strains used, and the methods of investigation. Based on the study of Kielbasa and the existing studies on germination and plant development [24,25,26,27,28,29,30,31,105], different MEL characteristics are likely to produce distinct effects. Due to the vast diversity and variety that we can find in nature, more research is needed to better understand the ways in which these complex mixtures of compounds, the EOs, act and how MEL might enhance or inhibit their bioactivity.

## 4. Materials and Methods

### 4.1. Selection of Essential Oils

For this study, three commercially available EOs were used: the cinnamon bark EO is native to Sri Lanka and was extracted from true cinnamon also known as Ceylon cinnamon (*Cinnamomum zeylanicum* Blume); the patchouli EO came from plants grown in Indonesia (*Pogostemon cablin* (Blanco) Benth.); and the geranium EO (*Pelargonium graveolens* L’Hér) was obtained from Africa and Madagascar.

### 4.2. Applied Treatments

The experimental design consisted of three exposures to a magnetic field, an electric field and laser irradiation, for each oil. Before each analysis, sample bottles of 2 mL containing 1 mL of the tested EOs were exposed to the investigated treatments as follows: the exposure to the magnetic field and the electric field was performed at USAMV Cluj-Napoca, while the exposure to the laser irradiation was performed at the National Institute for Research and Development of Isotopic and Molecular Technologies Cluj-Napoca.

The experiments were performed at room temperature at 20 ± 2 °C.

#### 4.2.1. Exposure to a Uniform Magnetic Field

The magnetic field was created by using Helmholtz coils that generated a uniform magnetic field of 0.22 × 10^−3^ T (G), which was measured with a Hall Sonde. Tested oils were exposed for 20 min [28].

#### 4.2.2. Exposure to a Uniform Electric Field

Samples were exposed to a homogenous electric field with an intensity of 158.2 V/m created by a parallel capacitator with a distance of 15.8 cm between the plates and a continuous supply voltage of 25 V. Exposure time was 20 min [28].

#### 4.2.3. Exposure to Laser Irradiation

The laser irradiation was a green light generated by a pulsed Nd:YAG Laser Quantum Gem at a wavelength of 532 nm and a frequency of 90 Hz generated by a Dsp Lock-In Amplifier Model Sr830 (Stanford Research System, Sunnyvale, CA, USA). Samples were exposed for 10 min.

### 4.3. GC-MS Analysis

The composition of the EOs was assessed by gas chromatography connected with mass spectrometry technique using a GC-MS Shimadzu model QP-2010 (Shimadzu Scientific Instruments, Kyoto, Japan), equipped with a Combi-PAL AOC-5000 (CTC Analytics, Zwingen, Switzerland) and a ZB- 5 ms capillary column (30 m × 0.25 mm i.d. × 0.25 µm film thickness, Phenomenex, Torrance, CA, USA).

From each sample, 1 µL of diluted hexane solution (1:50 split ratio) was transferred into the GC-MS injector. The volatile compounds were separated by following the following temperature program: from 50 °C (5 min) raised to 160 °C at a 5°/min rate, and then raised to 240 °C at a 10°/min rate and held at this temperature for 10 min. Helium was the carrier gas at a constant flow rate of 1 mL/min. The temperatures for the injector, ion source, and interface were set at 250 °C. The electron impact (EI) as an ion source was set at 70 eV for the MS detector that was operating in full scan (40–400 *m/z*). Using a minimum similarity of 85%, the recorded mass spectra were compared with those from the mass spectra libraries of the NIST27 and NIST147 software version 02, and the fragmentation patterns of the mass spectra were compared with database entries to facilitate the preliminary identification of the separated compounds. As a result, a qualitative evaluation of volatile chemicals was obtained, with each compound’s relative percentage assessed as a fraction of its integrated ion area from the 100% total ion chromatogram (TIC) area [56].

For the pictures of the structure of identified compounds, two databases were used: NIST Chemistry WebBook, SRD 69 and PubChem [119,120].

### 4.4. Antibacterial Activity

#### 4.4.1. Preparation of Microbial Strains

The following microorganisms were tested: *Staphylococcus aureus ATCC 6538P*, *Escherichia coli ATCC 25922*, *Salmonella enteritidis ATCC 13076*, *Listeria monocytogenes ATCC 19114*, *Pseudomonas aeruginosa ATCC 27853*, *Bacillus cereus ATCC 11778*, and *Candida albicans ATCC 10231*.

A tube containing 10 mL of sterile nutrient broth (Oxoid Ltd., Basingstoke, Hampshire, UK) maintained at 37 °C for 24 h, was used for the growth of each bacteria, except for *B. cereus* and *C. albicans*, which were grown at 30 °C, for 24 h. Additionally, a loopful of inoculum was inseminated on a selective medium: Palcam agar (Oxoid Ltd., Basingstoke, Hampshire, UK) for *L. monocytogenes*, Baird–Parker agar base supplemented with egg yolk tellurite emulsion for *S. aureus*, MYP agar supplemented with egg yolk emulsion and Polymyxin B (Oxoid Ltd.) for *B. cereus*, TBX agar for *E. coli*, Agar P (Oxoid Ltd.) for *P. aeruginosa*, XLD agar for *S. enteritidis* (Oxoid Ltd.), and YPD agar (Oxoid Ltd.) for *C. albicans* [121,122].

Thereafter, the plates were set into an incubator at 37 °C and 30 °C for *B. Cereus* and *C. Albicans*, for 24 h. Optical microscopy (Optika microscope, B-252, M.A.D.; Apparecchiature Scientifiche, Milan, Italy) was used to confirm the morphology and the purity of the inoculum. The turbidity of McFarland 0.5 standard (1.5 × 10^8^ CFU/mL) was matched by adjusting several colonies of standard cultures that were grown on Mueller–Hinton agar (Oxoid Ltd., Basingstoke, Hampshire, UK) at 37 °C for 24 h and on YPD agar at 30 °C for 24 h into a sterile saline solution (8.5 g/L) [122,123,124].

#### 4.4.2. Assessment of Minimum Inhibitory Concentration (MIC)

In order to calculate the MIC, the resazurin microtiter plate-based antibacterial assay was used. By combining eight parts ethanol (50%) and one part Tween 80, stock solutions of the EOs were made [123]. A 96-well microtiter plate was used, and 100 µL of sterile nutrient broth and 100 µL of sample were placed into the first well. A total of 100 µL was transferred from well to well (on row) in order to carry out serial 11-fold dilutions. From the last well of the row, 100 µL were discarded, and 10 µL of inoculum (1.5 × 10^8^ CFU/mL) were added to each well. As a positive control, Gentamicin (0.04 mg/mL in saline solution) was used, while for the negative control, one part of the saline solution, eight parts 50% ethanol, and one part Tween 80 were combined. For the next 20–22 h, microplates were incubated at 30 °C for *C. albicans* and *B. cereus* and at 37 °C for the rest of the bacterial strains. After the addition of 20 µL of resazurin aqueous solution (0.2 mg/mL) to each well, the microplates were incubated at 37 °C for 2 h, except for *B. cereus* and *C. albicans*, which were incubated for 2 h at 30 °C. The concentration at which the blue hue did not turn pink revealed the minimum inhibitory concentration (MIC). For every sample, three replicates were conducted [124].

#### 4.4.3. Assessment of the Minimum Bactericidal Concentration (MBC)

MBC was identified by plating a 10-μL aliquot from the final 4 wells that demonstrated bacterial growth inhibition in the MIC test, onto Mueller–Hinton and YPD Agar solid culture mediums (Oxoid Ltd., Basingstoke, Hampshire, UK). The plates were incubated for 24 h at 37 °C and at 30 °C, respectively. MBC was identified as the lowest concentration that stopped bacteria from growing and left no colonies on the plate. Three distinct biological replicates were carried out for every plate [122,125].

#### 4.4.4. Statistical Analysis for Microbiological Activity

The findings for the biological activity were presented as average mean ± standard deviation (SD) for determinations made in triplicate. To assess the average mean values and to evaluate the statistical differences, an ANOVA analysis of variance and Tukey’s honestly significant difference (HSD) test were chosen with a confidence interval of 95% or 99%. Results were processed through SPSS 19.0 statistical analysis (IBM, New York, NY, USA). A *p*-value was regarded as statistically significant if it was less than 0.05 [124].

### 4.5. Determination of Antioxidant Activity

#### 4.5.1. Assay for Radical Scavenging with DPPH

The DPPH technique was used to determine the antioxidant activity of the essential oils [107,108]. The ability of the essential oils to scavenge free radicals was determined using spectrophotometry and evaluated in relation to the effects on DPPH (CAS 1898-66-4, Sigma Aldrich GmbH, Steinheim, Germany) radical formation of standard solutions of methanol Trolox (6-hydroxy-2,5,7,8-tetramethylchroman-2-carboxylic acid, Tokyo Chemical Industry Co. Ltd., Tokyo, Japan) (0.025 mol/mL). The solutions were created as follows: The working solution was created by combining 90 mL of methanol with 29 mL of stock solution in order to obtain an absorbance of 1.1 ± 0.02 units, measured at 515 nm with a spectrophotometer. The blank sample was made by combining 150 µL of methanol with 2850 µL of DPPH solution. After 30 s of vortexing, the reactant mixture was allowed to react at room temperature for 60 min in the dark. The sample solutions were created by mixing 75 µL of essential oil and 75 µL of methanol with 2850 µL of working solution DPPH. After vortexing, they were allowed to rest in the dark for 60 min. Each sample’s absorbance was recorded at 515 nm against a blank of methanol. The Trolox calibration curve was utilized to quantify the antioxidant activity, which was then expressed in µM TE/mL sample.

#### 4.5.2. Assay for Radical Scavenging with ABTS^+^

The evaluation of the antioxidant activity of essential oils was examined by applying the ABTS^+^ method [107]. The methodology relies on the percentage inhibition in the peroxidation of this radical, observable at a wavelength of 734 nm, in an alkaline medium as a darkening of a blue-green solution. The stock solution was prepared by adding equal volumes (25 mL) of potassium persulfate solution (2.45 mM) and ABTS^+^ (7.4 mM) ([2,2′-Azino-bis(3-ethylbenzothiazoline-6-sulfonic acid (98%), Alfa Aesar, Karlsruhe, Germany, J65535). The resulting solution was then left in the dark (12 h) at room temperature to react. Following, the solution was diluted by adding 60 mL of methanol to 1 mL of ABTS^+^ solution until an absorbance of 1.1 ± 0.02 units was reached. The essential oil solutions (75 µL) were mixed with 75 µL methanol (Chempur, Piekary, Slavkie, Polonia) and 2850 µL of the ABTS^+^ solution in the dark for 2 h. Afterwards, the absorbance was measured at 734 nm. The standard Trolox [(R)-(+)-6-hydroxy-2,5,7,8-tetramethyl-chromal-2-carboxylic acid (98%), Sigma-Aldrich Chemie GmbH, Steinheim, Switzerland-391913] calibration was used. Results were expressed as µM Trolox equivalents (TE) per mL of sample.

## 5. Conclusions

In the current study, the influence of MEL on the EOs of cinnamon, patchouli, and geranium was investigated in terms of composition, antimicrobial properties, and antioxidant activity. The GC–MS analysis revealed 26 compounds for CEO with the prevalence of cinnamaldehyde, 17 for PEO with a majority of patchoulol, and 24 for GEO with (R)-citronellol as a major constituent. The investigation of the energized samples showed for CEO and GEO slight variations regarding the percentages and presence of minor compounds, while the profile of PEO was more stable. The investigation of MIC and MBC demonstrated a great antimicrobial activity in all oils on most microbial strains. Furthermore, in certain cases, several energized samples proved an enhancement of the antibacterial, bactericidal, and antifungal effects. Thus, we observed that the magnetic and electric exposures increased the antibacterial activity of all oils against *S. aureus*, while the laser irradiation treatment enhanced the antimicrobial activity of geranium EO in all cases, except for *S. enteritidis.* As the antifungal effect against *C. albicans* was boosted by all treatments on geranium EO, the exposure of cinnamon EO and patchouli EO to the electric field has substantially reduced their activity. Notably, the MIC for patchouli EO against *P. aeruginosa* has been improved by all exposures. While the magnetic field has increased the antimicrobial activity of patchouli EO against *L. monocytogenes*, for geranium EO, this effect was reduced. Additionally, this study reveals that the antioxidant activity was not changed by the applied treatments. The influences induced by the exposures to MEL varied depending on the oils, the microbial strains, and the methodology used to assess the effects.

The underlying mechanisms for these changes are not yet understood, and further research could bring new awareness regarding the improvement of the efficiency of EOs and their complex mechanisms of action and could help us find new solutions for the multiple problems we are facing, among which the menace of antimicrobial resistance is of top priority.

## Figures and Tables

**Table 1 plants-13-01992-t001:** Chemical composition of cinnamon bark (*Cinnamomum zeylanicum* Blume) essential oils.

Class	Compound Name	Structure	Percentage (%) from Total Peak Area
CEO	CEOM	CEOEl	CEOL
phenylpropanoid	Cinnamaldehyde	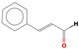	67.39	67.56	67.81	67.58
phenylpropanoid	Eugenol	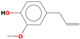	6.68	6.69	7.02	7.05
phenylpropanoid	Cinnamyl acetate	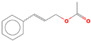	6.65	6.63	6.75	6.67
phenylpropanoid	Eugenol acetate	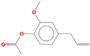	0.41	0.40	0.42	0.41
phenylpropanoids	Safrole	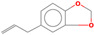	0.14	0.12	0.13	0.12
acyclic monoterpene	β-Linalool	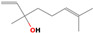	4.54	4.55	4.42	4.51
cyclic monoterpene	D-Limonene	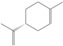	1.73	1.83	1.55	1.66
bicyclic monoterpene	α-Pinene	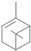	1.62	1.51	1.39	1.48
cyclic monoterpene	p-Cymene	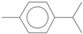	1.42	1.38	1.35	1.40
cyclic monoterpene	α-Phellandrene	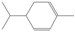	1.37	1.30	1.24	1.30
bicyclic monoterpene	Camphene	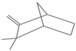	0.53	0.51	0.46	0.50
bicyclic monoterpene	β-Pinene	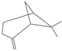	0.48	0.46	0.43	0.45
acyclic monoterpenoid	α-Terpineol	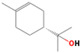	0.23	0.23	0.23	0.23
bicyclic monoterpene	α-Thujene	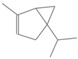	0.21	0.20	0.18	0.19
cyclic ether monoterpenoid	Eucalyptol	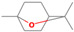	0.17	0.17	0.16	0.18
monoterpene	α-Terpinene	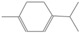	0.14	0.13	0.14	nd *
bicyclic monoterpene	δ-2-Carene	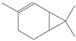	0.08	nd	nd	Nd
acyclic monoterpene	β-Myrcene	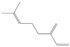	0.06	0.05	0.05	0.05
monoterpenoid alcoholic	1-Terpinen-4-ol	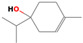	0.03	0.04	0.03	0.03
cyclic monoterpene	γ-Terpinene	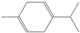	0.02	nd	nd	0.02
cyclic monoterpene	Terpinolene	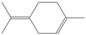	nd	0.08	0.07	0.08
acyclic monoterpene	β-*cis*-Ocimene	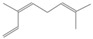	nd	0.12	0.10	0.11
bicyclic sesquiterpene	Caryophyllene	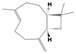	5.23	5.09	5.23	5.20
cyclic sesquiterpene	α-Caryophyllene	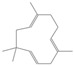	0.29	0.29	0.20	0.19
tricyclic sesquiterpene	Copaene	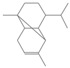	0.27	0.27	0.26	0.26
bicyclic sesquiterpenoid	Caryophyllene oxide	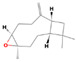	0.16	0.27	0.27	0.26
Other (Arene carbaldehyde)	Benzaldehyde	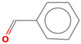	0.09	0.07	0.07	0.08
Other (alkyl carboxylic acid)	Butanoic acid, 2-methyl-, 2-methylbutyl ester	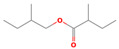	0.03	0.04	0.04	nd

* nd = not detected.

**Table 2 plants-13-01992-t002:** Chemical composition of patchouli (*Pogostemon cablin* (Blanco) Benth.) essential oils.

Class	Compound Name	Structure	Percentage (%) from Total Peak Area
PEO	PEOM	PEOEl	PEOL
tricyclic sesquiterpenoid	Patchoulol	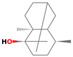	37.70	38.08	38.26	37.77
sesquiterpene	α-Bulnesene	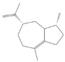	19.96	20.19	20.02	20.20
sesquiterpene	α-Guaiene	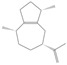	15.40	15.28	15.22	15.43
tricyclic sesquiterpene	Seychellene	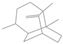	8.12	8.02	8.04	8.18
sesquiterpene	α-Patchoulene	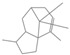	5.35	5.19	5.24	5.22
bicyclic sesquiterpene	Caryophyllene	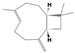	3.71	3.65	3.59	3.64
sesquiterpene	β-Patchoulene	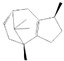	3.04	2.96	2.97	2.91
sesquiterpene	β-Elemene	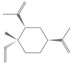	1.26	1.22	1.24	1.21
sesquiterpene	Patchoulene	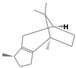	1.25	1.27	1.27	1.31
tricyclic sesquiterpene	Copaene	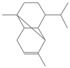	0.68	0.64	0.64	0.63
monocyclic sesquiterpene	α-Caryophyllene (humulene)	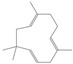	0.54	0.56	0.54	0.57
tricyclic sesquiterpene	(-)-α-Panasinsen	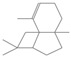	0.22	0.20	0.20	0.21
sesquiterpene	δ-Elemene	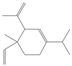	0.11	0.10	0.10	0.10
tricyclic sesquiterpene	α-Cubebene	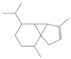	0.02	0.02	0.02	0.02
bicyclic monoterpene	β-Pinene	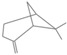	0.12	0.12	0.11	0.11
bicyclic monoterpene	α-Pinene	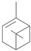	0.04	0.04	0.04	0.04
non-identified	non-identified		2.48	2.47	2.50	2.46

**Table 3 plants-13-01992-t003:** Chemical composition of geranium (*Pelargonium graveolens* L’Hér) essential oils.

Class	Compound Name	Structure	Percentage (%) from Total Peak Areaα
GEO	GEOM	GEOEl	GEOL
acyclic monoterpenoid	(R)-Citronellol	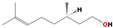	44.28	44.01	43.29	43.79
acyclic monoterpenoid	*cis*-Geraniol	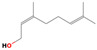	20.92	20.42	21.04	20.21
acyclic monoterpenoid	Citronellol acetate	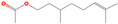	10.16	9.80	9.83	10.08
cyclic monoterpenoid	*cis*-Menthone	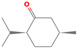	5.98	5.66	5.67	5.73
acyclic monoterpenoid	β-Linalool	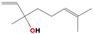	4.61	4.29	4.30	4.50
acyclic monoterpenoid	*cis*-Geranyl acetate	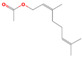	3.79	3.46	3.49	3.75
cyclic monoterpenoid	*Cis*-Rose-oxide	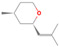	1.25	1.16	1.14	1.20
acyclic monoterpenoid	Geranial	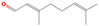	0.72	0.72	0.70	0.71
acyclic monoterpenoid	Citronellyl propionate	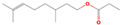	0.69	0.63	0.56	0.63
cyclic monoterpenoid	*trans*-Rose oxide	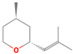	0.54	0.51	0.49	0.53
acyclic monoterpenoid	*cis*-Citral	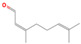	0.54	0.44	0.46	0.50
cyclic monoterpenoid	α-Terpineol	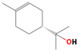	0.43	0.37	0.39	0.40
bicyclic monoterpene	α-Pinene	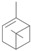	0.42	0.38	0.37	nd *
cyclic monoterpenoid	Menthol	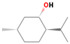	0.20	0.17	0.20	0.17
acyclic monoterpene	β-Myrcene	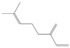	0.15	0.13	0.13	0.13
cyclic monoterpene	D-Limonene	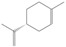	0.13	0.17	0.15	0.14
acyclic monoterpene	β-cis-Ocimene	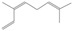	0.08	0.07	0.05	0.09
monocyclic monoterpene	p-Cymene	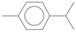	0.07	0.06	0.06	0.05
monocyclic monoterpene	α-Phellandrene	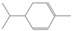	0.06	0.06	0.06	0.06
acyclic monoterpene	β-*trans*-Ocimene	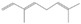	0.05	0.07	0.04	0.04
cyclic monoterpenoid	*trans*-Menthone	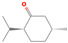	nd	1.96	1.95	1.96
acyclic monoterpenoid	(R)-(+)-Citronellal	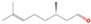	nd	0.11	0.11	0.09
cyclic monoterpenoid	Isopregol	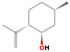	nd	0.10	0.10	0.11
bicyclic sesquiterpene	Caryophyllene	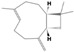	1.87	1.78	1.81	1.78
tricyclic sesquiterpene	β-Bourbonene	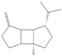	1.55	1.46	1.48	1.45
tricyclic sesquiterpene	Copaene	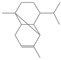	1.27	1.19	1.20	1.22
non-identified	non-identified		0.25	0.24	0.23	0.26
non-identified	non-identified		nd	0.66	0.68	nd

* nd = not detected.

**Table 4 plants-13-01992-t004:** Assessment of the minimum inhibitory concentration and the minimum bactericidal concentration values of cinnamon bark (*Cinnamomum zeylanicum* Blume) essential oils.

	*Gram* (*+*)						*Gram* (*-*)						*Fungi*	
Samples	*Staphylococcus aureus* *ATCC 6538P*		*Bacillus cereus* *ATCC 11778*		*Listeria monocytogenes* *ATCC 19114*		*Escherichia coli* *ATCC 25922*		*Salmonella enteritidis* *ATCC 13076*		*Pseudomonas* *aeruginosa* *ATCC 27853*		*Candida albicans* *ATCC 10231*	
	MIC (μL/mL)	MBC (μL/mL)	MIC (μL/mL)	MBC (μL/mL)	MIC (μL/mL)	MBC (μL/mL)	MIC (μL/mL)	MBC (μL/mL)	MIC (μL/mL)	MBC (μL/mL)	MIC (μL/mL)	MBC (μL/mL)	MIC (μL/mL)	MBC (μL/mL)
**CEO**	0.56 ± 0.00 ^b^	1.17 ± 0.00 ^b^	0.18 ± 0.08 ^c^	0.56 ± 0.00 ^c^	0.76 ± 0.35 ^b^	2.45 ± 0.00 ^a^	0.27 ± 0.00 ^c^	0.56 ± 0.00 ^b^	0.27 ± 0.00 ^c^	1.17 ± 0.00 ^b^	0.27 ± 0.00 ^d^	0.56 ± 0.00 ^d^	0.13 ± 0.00 ^c^	0.27 ± 0.00 ^d^
**CEOM**	0.22 ± 0.08 ^c^	0.56 ± 0.00 ^c^	0.46 ± 0.17 ^a^	0.97 ± 0.35 ^b^	0.97 ± 0.35 ^a^	2.45 ± 0.00 ^a^	0.97 ± 0.35 ^a^	4.24 ± 1.55 ^a^	0.97 ± 0.35 ^a^	4.24 ± 1.55 ^a^	2.45 ± 0.00 ^a^	5.14 ± 0.00 ^a^	0.18 ± 0.08 ^c^	0.37 ± 0.17 ^c^
**CEOEl**	0.27 ± 0.00 ^c^	0.56 ± 0.00 ^c^	0.22 ± 0.08 ^b^	0.56 ± 0.00 ^c^	0.56 ± 0.00 ^c^	2.45 ± 0.00 ^a^	0.37 ± 0.17 ^b^	0.56 ± 0.00 ^b^	0.22 ± 0.08 ^c^	0.56 ± 0.00 ^c^	0.56 ± 0.00 ^c^	1.17 ± 0.00 ^c^	30.99 ± 14.40 ^a^	47.62 ± 0.00 ^a^
**CEOL**	1.17 ± 0.00 ^a^	5.14 ± 0.00 ^a^	0.27 ± 0.00 ^b^	1.17 ± 0.00 ^a^	0.22 ± 0.08 ^d^	0.46 ± 0.17 ^b^	0.27 ± 0.00 ^c^	0.56 ± 0.00 ^b^	0.46 ± 0.17 ^b^	1.17 ± 0.00 ^b^	1.60 ± 0.74 ^b^	3.35 ± 1.55 ^b^	1.17 ± 0.00 ^b^	2.45 ± 0.00 ^b^
***p*** **value**	*p* < 0.001	*p* < 0.001	*p* < 0.001	*p* < 0.001	*p* < 0.001	*p* < 0.01	*p* < 0.001	*p* < 0.01	*p* < 0.001	*p* < 0.001	*p* < 0.001	*p* < 0.001	*p* < 0.001	*p* < 0.001
**Sig**	***	***	***	***	***	**	***	**	***	***	***	***	***	***

Note: Values are expressed as mean of three replicates. Values with different letters in the same column indicate statistically significant (Sig) differences (Tukey’s test), NS not significant, *p* > 0.05; ** very significant *p* ≤ 0.01; *** extremely significant *p* ≤ 0.001.

**Table 5 plants-13-01992-t005:** Assessment of the minimum inhibitory concentration and the minimum bactericidal concentrationvalues of patchouli (*Pogostemon cablin* (Blanco) Benth.) essential oils.

	*Gram* (*+*)						*Gram* (*-*)						*Fungi*	
Samples	*Staphylococcus aureus* *ATCC 6538P*		*Bacillus cereus* *ATCC 11778*		*Listeria monocytogenes* *ATCC 19114*		*Escherichia coli* *ATCC 25922*		*Salmonella enteritidis* *ATCC 13076*		*Pseudomonas* *aeruginosa* *ATCC 27853*		*Candida albicans* *ATCC 10231*	
	MIC (μL/mL)	MBC (μL/mL)	MIC (μL/mL)	MBC (μL/mL)	MIC (μL/mL)	MBC (μL/mL)	MIC (μL/mL)	MBC (μL/mL)	MIC (μL/mL)	MBC (μL/mL)	MIC (μL/mL)	MBC (μL/mL)	MIC (μL/mL)	MBC (μL/mL)
**PEO**	0.46 ± 0.17 ^b^	1.17 ± 0.00 ^b^	0.27 ± 0.00 ^c^	0.56 ± 0.00 ^b^	1.17 ± 0.00 ^a^	3.35 ± 1.55 ^a^	0.46 ± 0.17 ^d^	1.17 ± 0.00 ^d^	14.76 ± 6.86 ^a^	47.62 ± 0.00 ^a^	30.99 ± 14.40 ^a^	47.62 ± 0.00 ^a^	0.56 ± 0.00 ^c^	2.45 ± 0.00 ^d^
**PEOM**	0.27 ± 0.00 ^c^	0.56 ± 0.00 ^c^	0.27 ± 0.00 ^c^	0.56 ± 0.00 ^b^	0.27 ± 0.00 ^d^	0.27 ± 0.00 ^d^	2.45 ± 0.00 ^b^	10.80 ± 0.00 ^b^	10.80 ± 0.00 ^b^	22.68 ± 0.00 ^b^	14.76 ± 6.86 ^c^	30.99 ± 14.40 ^b^	2.45 ± 0.00 ^b^	5.14 ± 0.00 ^c^
**PEOEl**	0.13 ± 0.00 ^d^	0.27 ± 0.00 ^d^	1.17 ± 0.00 ^a^	3.35 ± 1.55 ^a^	0.76 ± 0.35 ^b^	2.45 ± 0.00 ^b^	30.99 ± 14.40 ^a^	47.62 ± 0.00 ^a^	8.91 ± 3.27 ^c^	22.68 ± 0.00 ^b^	22.68 ± 0.00 ^b^	47.62 ± 0.00 ^a^	8.91 ± 3.27 ^a^	22.68 ± 0.00 ^a^
**PEOL**	0.97 ± 0.35 ^a^	2.02 ± 0.74 ^a^	0.37 ± 0.17 ^b^	0.56 ± 0.00 ^b^	0.37 ± 0.17 ^c^	0.56 ± 0.00 ^c^	2.02 ± 0.74 ^c^	5.14 ± 0.00 ^c^	14.76 ± 6.86 ^a^	22.68 ± 0.00 ^b^	22.68 ± 0.00 ^b^	22.68 ± 0.00 ^c^	8.91 ± 3.27 ^a^	10.80 ± 0.00 ^b^
***p*** **value**	*p* < 0.001	*p* < 0.001	*p* < 0.001	*p* < 0.05	*p* < 0.001	*p* < 0.001	*p* < 0.001	*p* < 0.001	*p* < 0.001	*p* < 0.05	*p* < 0.001	*p* < 0.001	*p* < 0.001	*p* < 0.001
**Sig**	***	***	***	*	***	***	***	***	***	*	***	***	***	***

Note: Values are expressed as mean of three replicates. Values with different letters in the same column indicate statistically significant (Sig) differences (Tukey’s test), NS not significant, *p* > 0.05; * significant *p* ≤ 0.05; *** extremely significant *p* ≤ 0.001.

**Table 6 plants-13-01992-t006:** Assessment of the minimum inhibitory concentration and the minimum bactericidal concentration values of the geranium (*Pelargonium graveolens* L’Hér) essential oils.

	*Gram* (*+*)						*Gram* (*+*)						*Fungi*	
Samples	*Staphylococcus aureus* *ATCC 6538P*		*Bacillus cereus* *ATCC 11778*		*Listeria monocytogenes* *ATCC 19114*		*Escherichia coli* *ATCC 25922*		*Salmonella enteritidis* *ATCC 13076*		*Pseudomonas* *aeruginosa* *ATCC 27853*		*Candida albicans* *ATCC 10231*	
	MIC (μL/mL)	MBC (μL/mL)	MIC (μL/mL)	MBC (μL/mL)	MIC (μL/mL)	MBC (μL/mL)	MIC (μL/mL)	MBC (μL/mL)	MIC (μL/mL)	MBC (μL/mL)	MIC (μL/mL)	MBC (μL/mL)	MIC (μL/mL)	MBC (μL/mL)
**GEO**	2.45 ± 0.00 ^a^	5.14 ± 0.00 ^a^	0.56 ± 0.00 ^c^	2.45 ± 0.00 ^b^	1.60 ± 0.74 ^c^	5.14 ± 0.00 ^b^	0.46 ± 0.17 ^b^	0.97 ± 0.35 ^b^	0.56 ± 0.00 ^d^	1.17 ± 0.00 ^c^	5.14 ± 0.00 ^b^	10.8 ± 0.00 ^c^	0.56 ± 0.00 ^a^	2.45 ± 0.00 ^a^
**GEOM**	0.27 ± 0.00 ^d^	1.17 ± 0.00 ^c^	2.02 ± 0.74 ^a^	4.24 ± 1.55 ^a^	8.91 ± 3.27 ^a^	18.72 ± 6.86 ^a^	1.17 ± 0.00 ^a^	2.45 ± 0.00 ^a^	3.35 ± 1.55 ^a^	7.03 ± 3.27 ^a^	22.68 ± 0.00 ^a^	47.62 ± 0.00 ^a^	0.37 ± 0.17 ^b^	0.76 ± 0.35 ^c^
**GEOEl**	0.56 ± 0.00 ^c^	2.45 ± 0.00 ^b^	0.76 ± 0.35 ^b^	2.45 ± 0.00 ^b^	3.35 ± 1.55 ^b^	5.14 ± 0.00 ^b^	0.37 ± 0.17 ^c^	0.56 ± 0.00 ^c^	2.02 ± 0.74 ^c^	5.14 ± 0.00 ^b^	5.14 ± 0.00 ^b^	22.68 ± 0.00 ^b^	0.18 ± 0.08 ^d^	1.17 ± 0.00 ^b^
**GEOL**	0.76 ± 0.35 ^b^	2.45 ± 0.00 ^b^	0.22 ± 0.08 ^d^	0.27 ± 0.00 ^c^	0.56 ± 0.00 ^d^	1.17 ± 0.00 ^c^	0.27 ± 0.00 ^d^	0.56 ± 0.00 ^c^	2.45 ± 0.00 ^b^	5.14 ± 0.00 ^b^	3.35 ± 1.55 ^c^	10.8 ± 0.00 ^c^	0.22 ± 0.08 ^c^	0.56 ± 0.00 ^d^
***p*** **value**	*p* < 0.001	*p* < 0.001	*p* < 0.001	*p* < 0.001	*p* < 0.001	*p* < 0.001	*p* < 0.001	*p* < 0.001	*p* < 0.001	*p* < 0.001	*p* < 0.001	*p* < 0.001	*p* < 0.001	*p* < 0.001
**Sig**	***	***	***	***	***	***	***	***	***	***	***	***	***	***

Note: Values are expressed as mean of three replicates. Values with different letters in the same column indicate statistically significant (Sig) differences (Tukey’s test), NS not significant, *p* > 0.05; *** extremely significant *p* ≤ 0.001.

**Table 7 plants-13-01992-t007:** Antioxidant activity of cinnamon bark (*Cinnamomum zeylanicum* Blume) essential oils.

Botanical Family	Sample	DPPH	ABTS^+^
		[µM TE/mL]	
Lauraceae	**CEO**	1.86 ± 0.35 ^a^	2.19 ± 0.17 ^a^
	**CEOM**	1.86 ± 0.35 ^a^	2.19 ± 0.17 ^a^
	**CEOEl**	1.86 ± 0.35 ^a^	2.19 ± 0.17 ^a^
	**CEOL**	1.86 ± 0.35 ^a^	2.19 ± 0.17 ^a^
	***p*** **value**	*p* > 0.05	*p* > 0.05
	**Sig**	NS	NS

Note: Values are expressed as mean of three replicates. Values with different letters in the same column indicate statistically significant (Sig) differences (Tukey’s test), NS not significant, *p* > 0.05.

**Table 8 plants-13-01992-t008:** Antioxidant activity of patchouli (*Pogostemon cablin* (Blanco) Benth.) essential oils.

Botanical Family	Sample	DPPH	ABTS^+^
		[µM TE/mL]	
Lamiaceae	**PEO**	1.85 ± 0.74 ^a^	2.18 ± 0.005 ^a^
	**PEOM**	1.85 ± 0.74 ^a^	2.18 ± 0.005 ^a^
	**PEOEl**	1.85 ± 0.74 ^a^	2.18 ± 0.005 ^a^
	**PEOL**	1.85 ± 0.74 ^a^	2.18 ± 0.005 ^a^
	***p*** **value**	*p* > 0.05	*p* > 0.05
	**Sig**	NS	NS

Note: Values are expressed as mean of three replicates. Values with different letters in the same column indicate statistically significant (Sig) differences (Tukey’s test), NS not significant, *p* > 0.05.

**Table 9 plants-13-01992-t009:** Antioxidant activity of geranium (*Pelargonium graveolens* L’Hér) essential oils.

Botanical Family	Sample	DPPH	ABTS^+^
		[µM TE/mL]	
Geraniaceae	**GEO**	1.85 ± 0.74 ^a^	2.18 ± 0.005 ^a^
	**GEOM**	1.85 ± 0.74 ^a^	2.18 ± 0.005 ^a^
	**GEOEl**	1.85 ± 0.74 ^a^	2.18 ± 0.005 ^a^
	**GEOL**	1.85 ± 0.74 ^a^	2.18 ± 0.005 ^a^
	***p*** **value**	*p* > 0.05	*p* > 0.05
	**Sig**	NS	NS

Note: Values are expressed as mean of three replicates. Values with different letters in the same column indicate statistically significant (Sig) differences (Tukey’s test), NS not significant, *p* > 0.05.

## Data Availability

Data from this study are available in the text of the paper.

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
