# Peer review of "The Influence of Physical Fields (Magnetic and Electric) and LASER Exposure on the Composition and Bioactivity of Cinnamon Bark, Patchouli, and Geranium Essential Oils"

_plants, 2024, doi:10.3390/plants13141992_

Round 1

Reviewer 1 Report

Comments and Suggestions for Authors

Scheau and colleagues investigated the antimicrobial and antioxidant activity of three essential oils which were previously exposed to physical fields or to a laser. The analytical comparison of the composition of the original and the treated essential oils were compared. In conclusion only slight differences regarding the ingredients were observed in the treated samples, but an improvement of antimicrobial activity against certain bacterial and antifungal strains was determined.

The authors have selected a most interesting topic as there is an urgent need for new antimicrobial substances and treatment options. The whole study was planned and performed in an excellent way and the manuscript is written in a fluid style. Altogether 125 references where cited in the manuscript.

HOWEVER, there are some issues which should be reconsidered by the authors as follows:

1.    The botanical name should be corrected throughout the whole manuscript: „Pelargonium graveolens“. Please correct this plant name!

2.    Page 3, when the plants are characterized reagarding their habitus and geographical origin (the three paragraphs starting with „Cinnamomum EO is extracted …“): In scientific papers dealing with plants the full botanical name including the author name should be presented when the plant is introduced to the readers. Therefore please change „Cinnamomum zeylanicum“ to „Cinnamomum zeylanicum Blume“, „Pogostemon cablin“ to „Pogostemon cablin (Blanco) Benth.“ and „Pelargonium graveolens“ to „Pelargonium graveolens L'Hér.“ In recent years the botanical name „Cinnamomun zeylanicum“ has more or less been replaced by „Cinnamomum verum“ and „Cinnamomum verum J.Presl“, respectively, thus the latter botanical name should at least be mentioned in brackets regarding the Cinnamon plant. Please add this synonym!

3.    In chapter 2.1.1 the analytical method (GC-MS) and the method of quantification needs to be mentioned in brief. It is not sufficient to describe the method and quantification in the material and methods chapter. Please add this information!

4.    In addition chromatograms of the three untreated essential oils should be presented together with a legend (assignment of the peaks in the chromatogram to distinct compounds) for the readership. These chromatograms are of some importance in order to evaluate the separation of the components of the essential oils. Please consider whether it is appropriate to enclose the chromatograms and the assignments in the supplementary part, preferably before the list of abbreviations!

5.    In Tables 1, 2 and 3 the retention times and also the main mass peak(s) and prominent mass fragments need to be added in the tables as the compounds were detected by mass spectroscopy. Please add this important information!

6.    In chapter 2.1.2 on patchouli and Table 2 the compound should be named either patchoulol or patchouli alcohol, but in a consistent way!
7.    Please change „antifungical“ to „antifungal“ throughout the whole manuscript!

8.    On page 13 of 23, in the paragraph beginning with „In several studies, CEO proved ….“ please add in the sentence starting with „Various IC50 were reported“ the test assays applied for the determination of antioxidant activities!

9.    On page 14 of 23, in the paragraph beginning with „In a wider study on 42 EOs, …“ the concentrations for the values obtained in reference 109 needs to be added.

10.    In Chapter 4 / Material and methods, the references and/or chemicals (e.g. Trolox, ABTS, DPPH) and their origin (e.g. Sigma-Aldrich, Germany) need to be added. In addition the name of the company (producer of the essential oil), the batches and if possible (and known) also the geographical origin of the essential oils should be disclosed to the readership. Please add this relevant information as the essential oils from other (=different) geographical origins might differ in terms of their composition.

11.    Please write all botanical names in the references in italics!

To sum up, this is very good and informative study with scientific and potential clinical significance. Thus the results will be of great interest for the international readership of the journal. However, the issues mentioned should be addressed by the authors.

Author Response

Dear Reviewer,

The authors would like to express their profound gratitude for taking the time to revise our manuscript and for your valuable suggestions which improved our paper.

On behalf of all authors, I re-submit the revised version of our manuscript, which includes all the suggestions received from you, as follows:

General Comment: “Scheau and colleagues investigated the antimicrobial and antioxidant activity of three essential oils which were previously exposed to physical fields or to a laser. The analytical comparison of the composition of the original and the treated essential oils were compared. In conclusion only slight differences regarding the ingredients were observed in the treated samples, but an improvement of antimicrobial activity against certain bacterial and antifungal strains was determined.

The authors have selected a most interesting topic as there is an urgent need for new antimicrobial substances and treatment options. The whole study was planned and performed in an excellent way and the manuscript is written in a fluid style. Altogether 125 references were cited in the manuscript.”

Answer: The authors would like to thank you for your positive appreciation of our work and for finding it important for the current research.

Comment (1): “The botanical name should be corrected throughout the whole manuscript: „Pelargonium graveolens“. Please correct this plant name!”

Answer: Thank you for your observation. We have performed the change in the revised manuscript.

Comment (2): “Page 3, when the plants are characterized regarding their habitus and geographical origin (the three paragraphs starting with „Cinnamomum EO is extracted …“): In scientific papers dealing with plants the full botanical name including the author name should be presented when the plant is introduced to the readers. Therefore please change „Cinnamomum zeylanicum“ to „Cinnamomum zeylanicum Blume“, „Pogostemon cablin“ to „Pogostemon cablin (Blanco) Benth.“ and „Pelargonium graveolens“ to „Pelargonium graveolens L'Hér.“ In recent years the botanical name „Cinnamomun zeylanicum“ has more or less been replaced by „Cinnamomum verum“ and „Cinnamomum verum J.Presl“, respectively, thus the latter botanical name should at least be mentioned in brackets regarding the Cinnamon plant. Please add this synonym!”

Answer: Thank you very much for your suggestion. We agree with your comment and we’ve performed the changes for all the Latin names throughout the manuscript.

Comment (3): “In chapter 2.1.1 the analytical method (GC-MS) and the method of quantification needs to be mentioned in brief. It is not sufficient to describe the method and quantification in the material and methods chapter. Please add this information!”

Answer: Thank you for your suggestion. The method was described in the 4.3 section and several more details were added according to your recommendation.

Comment (4): “In addition chromatograms of the three untreated essential oils should be presented together with a legend (assignment of the peaks in the chromatogram to distinct compounds) for the readership. These chromatograms are of some importance in order to evaluate the separation of the components of the essential oils. Please consider whether it is appropriate to enclose the chromatograms and the assignments in the supplementary part, preferably before the list of abbreviations!”

Answer: The authors appreciate your suggestion and all the chromatograms were included in the supplementary part of the article.

Comment (5): “In Tables 1, 2 and 3 the retention times and also the main mass peak(s) and prominent mass fragments need to be added in the tables as the compounds were detected by mass spectroscopy. Please add this important information!”

Answer: Thank you very much for making this suggestion. We agree that this information is important and, in order to keep the table more concise and also integrate other received comments by other reviewers, we have included the requested information in the supplementary materials section, in the Excel file: ”Chromatographic Data for Cinnamon, Patchouli and Geranium Essential Oils”, file. 

Comment (6): “In chapter 2.1.2 on patchouli and Table 2 the compound should be named either patchoulol or patchouli alcohol, but in a consistent way!”

Answer: We agree with your observation and we’ve corrected the manuscript.

Comment (7): “Please change „antifungical“ to „antifungal“ throughout the whole manuscript!”

Answer: Thank you for pointing this out, the changes have been made.

Comment (8): “On page 13 of 23, in the paragraph beginning with „In several studies, CEO proved ….“ please add in the sentence starting with „Various IC50 were reported“ the test assays applied for the determination of antioxidant activities!”

Answer: Thank you for your suggestions. We have introduced the antioxidant test at line 401: “Various IC50 for DPPH scavenging activity by cinnamon EOs were reported”

Comment (9): “On page 14 of 23, in the paragraph beginning with „In a wider study on 42 EOs, …“ the concentrations for the values obtained in reference 109 needs to be added.”

Answer: Thank you for your observation. We performed the change in the revised manuscript, between lines 417- 419, as follows: ”In a wider study on 42 EOs, the antioxidant activity on DPPH, at the concentration of 5 mg/mL, of CEO was found to be 91.4 % ± 0.002, while for the PEO it was 15.63% ± 0.009, which is contradictory to our results in regards to PEO [109].”

Comment (10): “In Chapter 4 / Material and methods, the references and/or chemicals (e.g. Trolox, ABTS, DPPH) and their origin (e.g. Sigma-Aldrich, Germany) need to be added. In addition, the name of the company (producer of the essential oil), the batches and if possible (and known) also the geographical origin of the essential oils should be disclosed to the readership. Please add this relevant information as the essential oils from other (=different) geographical origins might differ in terms of their composition.”

Answer: Thank you for your suggestions. We have added the origin of the chemicals used to determine the antioxidant activity of the essential oils. Additionally, we have used three commercially available EOs the cinnamon bark EO is native from Sri Lanka and is extracted from true cinnamon (Ceylon cinnamon - Cinnamomum zeylanicum Blume), patchouli EO comes from the plants grown in Indonesia (Pogostemon cablin (Blanco) Benth.) and geranium EO (Pelargonium graveolens L’Hér) is obtained from Africa and Madagascar.

Comment (11): “Please write all botanical names in the references in italics!”

Answer: Thank you for observing this, we have modified the names accordingly.

Last comment: “To sum up, this is very good and informative study with scientific and potential clinical significance. Thus, the results will be of great interest for the international readership of the journal. However, the issues mentioned should be addressed by the authors.”

Answer: We really appreciate your positive feedback and your acknowledgement of our work. Your suggestions were highly valued and have helped us to refine our article. Thank you!

Kind regards,

Camelia Scheau

Reviewer 2 Report

Comments and Suggestions for Authors

In this study reported that the Influence of Physical Fields (Magnetic and Electric) and LASER exposure on the Composition and Bioactivity of Cinna-mon Bark, Patchouli and Geranium Essential Oils. This article has made some contributions to in terms of essential oil utilization , but it is currently not suitable for acceptance and requires the following revisions:

1. Why not study the changes in essential oils after physical fields (magnetic and electric) and laser exposure to plants?

2. In fact, this study only used physical fields (magnetic and electric) and laser exposure of essential oils, resulting in changes in essential oils and biological activity? So, the introduction section needs to be modified? Emphasis should be placed on describing the effects of physical fields (magnetic and electric fields) and lasers on essential oils or other substances?

3. The discussion section is cumbersome and the hierarchy is unclear, and the structure needs to be adjusted. Emphasis should be placed on the possible mechanisms underlying changes in physical fields (magnetic and electric) and laser of essential oils, chemical composition, and biological activity?

4. After being exposed to physical fields (magnetic and electric fields) and laser irradiation for 20 minutes, the chemical composition and biological activity of essential oils changes. Essential oils exposed to physical fields (magnetic and electric fields) and lasers. So, How long can the effects of changes in composition and activity last?

Author Response

Dear Reviewer,

The authors would like to express their gratitude for taking the time to revise our manuscript and for your valuable suggestions which improved our paper.

On behalf of all authors, I re-submit the revised version of our manuscript, which includes all the suggestions received from you, as follows:

Comment (1): “In this study reported that the Influence of Physical Fields (Magnetic and Electric) and LASER exposure on the Composition and Bioactivity of Cinnamon Bark, Patchouli and Geranium Essential Oils. This article has made some contributions to in terms of essential oil utilization, but it is currently not suitable for acceptance and requires the following revisions:”

Answer: The authors would like to thank you for your appreciation regarding our manuscript.

Comment (2): “Why not study the changes in essential oils after physical fields (magnetic and electric) and laser exposure to plants?”

Answer: Thank you for your question. While there are more studies investigating the effects of physical fields (magnetic and electric) and laser exposure on plants, there is only one study that researched the effects of the magnetic field when applied directly on the essential oils. Therefore, the aim of our study was to investigate the influence of these treatments on the essential oils when it comes to chemical composition and bioactivities.

Comment (3): “In fact, this study only used physical fields (magnetic and electric) and laser exposure of essential oils, resulting in changes in essential oils and biological activity? So, the introduction section needs to be modified? Emphasis should be placed on describing the effects of physical fields (magnetic and electric fields) and lasers (MEL) on essential oils or other substances?”

Answer: Thank you very much for your suggestion. As the researched area of study is new, in the introduction we’ve presented the actual knowledge of the effects of these treatments on seeds and plants. Currently, there is limited information regarding the effects of MEL on essential oils and other substances (one similar study identified by us, which was presented in the discussion section, please refer to article 104). This has also inspired us to perform this study.

Comment (4): “The discussion section is cumbersome and the hierarchy is unclear, and the structure needs to be adjusted. Emphasis should be placed on the possible mechanisms underlying changes in physical fields (magnetic and electric) and laser of essential oils, chemical composition, and biological activity?”

Answer: Thank you for your feedback. The length of the discussion part is a consequence of the fact that EOs are complex mixtures of substances, that we’ve used three oils and three treatments, and that we’ve followed more parameters for this study. We have presented the data according to the structure presented in the Results section.

Comment (5): “After being exposed to physical fields (magnetic and electric fields) and laser irradiation for 20 minutes, the chemical composition and biological activity of essential oils changes. Essential oils exposed to physical fields (magnetic and electric fields) and lasers. So, How long can the effects of changes in composition and activity last?”

Answer: Thank you for your question. In this study we have evaluated the influence of an exposure of EOs to physical fields for 20 min. and to LASER irradiation for 10 min. in terms of chemical composition and bioactivities as presented in the article. While it is possible that the effects of these influences to change over time, these variations were not aligned with the purpose of this research and were not examined. This could be a valuable idea for future investigations.

Sincerely,

Camelia Scheau

Reviewer 3 Report

Comments and Suggestions for Authors

The present study aims to investigate the impact of magnetic, electric and laser (MEL) exposure on the composition and the antibacterial, antifungal, and antioxidant activity of three commercial EOs (cinnamon bark, patchouli, and geranium), well known for their antibacterial effect. The results obtained showed that the magnetic influence has improved the potency of patchouli EO against L. monocytogenes, S. enteritidis, and P. aeruginosa. The antimicrobial activity of cinnamon EO against L. monocytogenes was improved by electric and laser treatments. All exposures have increased the antifungical effect of geranium EO against C. albicans while the antioxidant activity was not modified by any of the treatments.

Generally, the manuscript is clear and presented in a well-structured manner. The English language is understandable. The majority of references are relevant and published in the last 5 years. There are not an excessive number of self-citations. The experimental design is appropriate and the results are reproducible based on the details given in the methods section. The statistical analysis is appropriate.

However,

Ø  In my opinion, the authors should provide the GC/MS chromatograms of the EOs studied and numbered the peaks according to the order of their elution from the column. The numbers given in the chromatograms should correspond to the numbers that should be added in Tables 1-3.

Ø  Also, a row “Content, %” should be inserted above the abbreviations (CEO, CEOM..) in Tables 1-3

Ø  Regarding the tables 4-9, the numbered values should be separated by the dot not by comma. There is not a positive control (antibiotics, antifungal agents) given in the Tables to which the obtained results could be compared.  

Ø  The terms cis-, trans-, R- describing the isomers, as well as m/z should be italicized throughout the whole manuscript

Ø  The authors should not use the term "concentration" for the components present in the EOs studied because they used the area normalization and the results represent the content expressed in percentages. Concentrations could be obtained by an internal standard, external standard or standard addition methods.

In conclusion, modification of the commercial EOs’ activity, which are, as the authors stated, well known for their antimicrobial activity, demands additional costs and equipment. The findings are significant in the "battle" against S. aureus, where the authors observed that magnetic and electric exposures increased the antibacterial activity of all oils against this microorganism, while laser irradiation treatment enhanced the antimicrobial activity of geranium EO in all cases except for S. enteritidis. As the antifungal effect against C. albicans was enhanced by all treatments on geranium EO, the exposure of cinnamon EO and patchouli EO to the electric field has substantially reduced their activity. Also, the treatments used didn’t affect the antioxidant activity of the studied EOs. The underlying mechanisms for the changes observed are lacking, so I am not sure that the conclusions are interesting for the readership of the journal because the paper doesn’t advance current knowledge.

Author Response

Dear Reviewer,

The authors would like to express their gratitude for taking the time to revise our manuscript and for your valuable suggestions which improved our paper.

On behalf of all authors, I re-submit the revised version of our manuscript, which includes all the suggestions received from you, as follows:

Comment (1): “The present study aims to investigate the impact of magnetic, electric and laser (MEL) exposure on the composition and the antibacterial, antifungal, and antioxidant activity of three commercial EOs (cinnamon bark, patchouli, and geranium), well known for their antibacterial effect. The results obtained showed that the magnetic influence has improved the potency of patchouli EO against L. monocytogenes, S. enteritidis, and P. aeruginosa. The antimicrobial activity of cinnamon EO against L. monocytogenes was improved by electric and laser treatments. All exposures have increased the antifungal effect of geranium EO against C. albicans while the antioxidant activity was not modified by any of the treatments.

Generally, the manuscript is clear and presented in a well-structured manner. The English language is understandable. The majority of references are relevant and published in the last 5 years. There are not an excessive number of self-citations. The experimental design is appropriate and the results are reproducible based on the details given in the methods section. The statistical analysis is appropriate.”

Answer: The authors would like to thank you for your appreciation regarding our manuscript.

Comment (2): “In my opinion, the authors should provide the GC/MS chromatograms of the EOs studied and numbered the peaks according to the order of their elution from the column. The numbers given in the chromatograms should correspond to the numbers that should be added in Tables 1-3.”

Answer: We would like to thank you for your suggestion which was taken into consideration. As we’ve searched to integrate as many comments by all the reviewers, we have decided to organize the first three tables according to the classes of the compounds. Additionally, two more files were added in the supplementary materials: one Word document “Chromatograms of the Essential Oils” with the chromatographs of the EOs and one Excel file: “Chromatographic Data for Cinnamon, Patchouli and Geranium Essential Oils” containing more details regarding the chromatographs, which orders the molecules based on their retention time. We hope this is suitable.

Comment (3): “Also, a row “Content, %” should be inserted above the abbreviations (CEO, CEOM..) in Tables 1-3.”

Answer: Thank you very much for your suggestion. The changes were performed the changes in the revised manuscript.

Comment (4): “Regarding the tables 4-9, the numbered values should be separated by the dot not by comma. There is not a positive control (antibiotics, antifungal agents) given in the Tables to which the obtained results could be compared.”

Answer: Thank you for your observation, the changes have been made in the tables. This study aimed to investigate the influence of the three treatments (magnetic and electric fields, and laser irradiation) on the composition and bioactivity of the three EOs, and therefore, in the design of this study, the reference used was the untreated EO for each group. 

Comment (5): “The terms cis-, trans-, R- describing the isomers, as well as m/z should be italicized throughout the whole manuscript.”

Answer: The authors agree with your comment and we performed the changes in the revised manuscript.

Comment (6): “The authors should not use the term "concentration" for the components present in the EOs studied because they used the area normalization and the results represent the content expressed in percentages. Concentrations could be obtained by an internal standard, external standard or standard addition methods.”

Answer: Thank you very much for your suggestion, we have corrected this aspect.

Comment (7): “In conclusion, modification of the commercial EOs’ activity, which are, as the authors stated, well known for their antimicrobial activity, demands additional costs and equipment. The findings are significant in the "battle" against S. aureus, where the authors observed that magnetic and electric exposures increased the antibacterial activity of all oils against this microorganism, while laser irradiation treatment enhanced the antimicrobial activity of geranium EO in all cases except for S. enteritidis. As the antifungal effect against C. albicans was enhanced by all treatments on geranium EO, the exposure of cinnamon EO and patchouli EO to the electric field has substantially reduced their activity. Also, the treatments used didn’t affect the antioxidant activity of the studied EOs. The underlying mechanisms for the changes observed are lacking, so I am not sure that the conclusions are interesting for the readership of the journal because the paper doesn’t advance current knowledge.”

Answer: Thank you for your observation. Since this is a highly new area of research, which has just started to be investigated (according to our knowledge there is only 1 study that reported the antimicrobial effects of the EOs pretreated in a magnetic field, please see reference 104) and taking into account that EOs present a complex composition with a large number of various molecules involved, the aim of this article was to report the observed modifications induced by the applied treatments and not to explain the underlying mechanisms, which, at this moment, have not yet been elucidated. As the chromatographs show, the changes in the composition of the EOs exposed are rather insignificant to support the changes in the antimicrobial effects, which leads us to believe that other, more subtle mechanisms are responsible for these effects. The novelty part of this research is to reveal that the antimicrobial effects of EOs can be modified and even optimized through these or similar treatments and thus, EOs can become even more powerful agents in the fight against bacterial and fungal pathologies. 

Sincerely,

Camelia Scheau

Reviewer 4 Report

Comments and Suggestions for Authors

This manuscript reports the influence of three treatments on the essential oils of three plant materials, focusing on their composition, antimicrobial properties, and antioxidant activities. The content of this article is extensive and intriguing. Below are the suggested improvements:

 1.          In section 2.1.1, "CEOM" and "CEOI" should be written as "CEOM and CEOI."

2.          To facilitate the comparison of compound changes between different types, it is suggested to rearrange Tables 1-3 by compound types. This will make it easier to reveal and discuss the relationship between changes in compounds and their bioactivity.

3.          In the notes for Tables 4-9, "significant (S)" should be written as "significant (Sig)." Additionally, each table should have its own specific note. For example, Table 4 does not need "NS." Tables 7-9 do not need "P>0.05; *significant P≤0.05; **very significant P≤0.01; ***extremely significant P≤0.001."

4.          The "+" in "ABTS+" should be superscripted.

5.          The discussion should address how the components of the essential oils are altered by the MEL treatment, explaining the possible reasons for these changes.

6.          In section 4.2, the sample treatment descriptions are not clear. In fact, each sample underwent a specific treatment. Please provide a clear explanation.

Comments on the Quality of English Language

English language required in the result sections.

Author Response

Dear Reviewer,

The authors would like to express their gratitude for taking the time to revise our manuscript and for your valuable suggestions which improved our paper.

On behalf of all authors, I re-submit the revised version of our manuscript, which includes all the suggestions received from you, as follows:

Comment (1): “This manuscript reports the influence of three treatments on the essential oils of three plant materials, focusing on their composition, antimicrobial properties, and antioxidant activities. The content of this article is extensive and intriguing.”

Answer: The authors would like to thank you for your appreciation regarding our manuscript.

Comment (2): “In section 2.1.1, "CEOM" and "CEOI" should be written as "CEOM and CEOI."

Answer: We agree with your suggestion and we performed the change in the revised manuscript.

Comment (3): “To facilitate the comparison of compound changes between different types, it is suggested to rearrange Tables 1-3 by compound types. This will make it easier to reveal and discuss the relationship between changes in compounds and their bioactivity.”

Answer: Thank you very much for your suggestion. We have rearranged the tables according to your recommendation.

Comment (4): “The "+" in "ABTS+" should be superscripted.”

Answer:  Thank you for your observation, the rectification has been implemented in the revised manuscript.

Comment (5): “The discussion should address how the components of the essential oils are altered by the MEL treatment, explaining the possible reasons for these changes.”

Answer: Thank you for your suggestion. As this is a new area of research with limited existing studies (we have found just 1 article that treated two EOs in a magnetic field – 104), our study aimed to report the influences of the three treatments on the three essential oils in terms of composition and bioactivity. Since the chromatographic analyses performed show insignificant changes at the level of the composition of the EOs to support the changes in the antimicrobial effects induced by the treatments, we suppose that other, more subtle mechanisms might be in charge, that have not yet been understood. We consider that further investigations are needed in order to reveal the mechanisms of action behind these induced effects.   

Comment (6): “In section 4.2, the sample treatment descriptions are not clear. In fact, each sample underwent a specific treatment. Please provide a clear explanation.”

Answer: Thank you very much for your observation. A sample from each oil was exposed to the three treatments as explained in section 4.2. and then compared with a control sample, represented by the untreated essential oil, for all the investigations performed.

Sincerely,

Camelia Scheau

Reviewer 5 Report

Comments and Suggestions for Authors

The authors presented composition and Bioactivity of cinnamon bark, patchouli and geranium essential oils, with GC-MS characterization followed by antimicrobial activity and antioxidant activity assays.

General comment:

The manuscript needs to be improved. The references are recent but not correctly reported. Too much self-references detected. Everything should be improved. Data Availability Statement correctly reported.  At present needs some improvements.

Specific comment:

Lines are not reported. However here some of my comments:

Abstract: “In recent years, essential oils (EOs) have received increased attention from the research community due to their potent and wide-ranging effects, including antibacterial, antifungal, antivi-ral, antioxidant, anti-inflammatory, sedative, anxiolytic, analgesic, and antiseptic properties.” Is too general. The introduction of the abstract should focus of Cinna-mon Bark, Patchouli and Geranium Essential Oils.

“Results show that the magnetic influence has improved the potency of patchouli EO against L. monocytogenes, S. enteritidis and P. aeruginosa” sometimes the past or the present is used in the same sentence.

Introduction: the introduction in too general in the first paragraph, it should be changed adding more detailed references on Cinnamon EO, Geranium (Pelagornium graveleons), EO of patchouli.

Too much references together, for example [5-9]. The authors should include maximum 3 references in the same place

“When considering the impact of the electric field, a study from 2021” it is not clear whether it refers to the authors' previous work or some other work.

What is the novelty of the work compared to previous similar studies?

Results:

Alpha and beta with symbols

The real statistical differences between such close numbers are not clear. For example for 1 compound in cinnamon bark (Cinnamon zeylanicum) essential oils, 67.39% 67.56% 67.81% 67.58% how different are them?

Antimicrobial activity: which and how many works have previously investigated the activity of these essential oils on these bacteria?

Table 4 is not completely visible

Discussion: too long and difficult to follow.

Materials and methods:For this study were used EOs of cinnamon bark (Cinnamomum zeylanicum), patchouli (Pogostemon cablin) and geranium (Pelagornium graveleons) which were acquired from the same international company.” Which company? More details are needed.

Antibacterial Activity is not correct if we talk about Candida albicans

Comments on the Quality of English Language

Moderate editing of English language required

Author Response

Dear Reviewer,

The authors would like to express their gratitude for taking the time to revise our manuscript and for your valuable suggestions which improved our paper.

On behalf of all authors, I re-submit the revised version of our manuscript, which includes all the suggestions received from you, as follows:

General comment: “The manuscript needs to be improved. The references are recent but not correctly reported. Too much self-references detected. Everything should be improved. Data Availability Statement correctly reported.  At present needs some improvements.”

Answer: Thank you for your suggestions, we are open to improve our manuscript and implement your suggestions.

Comment (1): “Lines are not reported. However here some of my comments.”

Answer:  Thank you for your observation, the lines were added.

Comment (2): “In recent years, essential oils (EOs) have received increased attention from the research community due to their potent and wide-ranging effects, including antibacterial, antifungal, antiviral, antioxidant, anti-inflammatory, sedative, anxiolytic, analgesic, and antiseptic properties.” Is too general. The introduction of the abstract should focus of Cinnamon Bark, Patchouli and Geranium Essential Oils.”

Answer: Thank you for your suggestion. The abstract part was revised accordingly. “In recent years, essential oils (EOs) have received increased attention from the research community and the EOs of cinnamon, patchouli and geranium have been highly recognized for their antibac-terial, antifungal, antiviral and antioxidant effects. Due to these properties, they become valuable, promising candidates to address the worldwide threat of antimicrobial resistance and of other diseases, as well.”

Comment (3): “Results show that the magnetic influence has improved the potency of patchouli EO against L. monocytogenes, S. enteritidis and P. aeruginosa” sometimes the past or the present is used in the same sentence.”

Answer: Thank you for your observation, the manuscript was revised from this point of view.

Comment (4): “Introduction: the introduction in too general in the first paragraph, it should be changed adding more detailed references on Cinnamon EO, Geranium (Pelagornium graveleons), EO of patchouli.”

Answer: Thank you for your suggestion. Given the novelty of this work, we chose to start with a more general approach and then go deeper into our thematic. The detailed description of the investigated EOs is presented from line 109 to 121.

Comment (5): “Too much references together, for example [5-9]. The authors should include maximum 3 references in the same place.”

Answer: Thank you very much for your suggestion, the manuscript was revised.

Comment (6): “When considering the impact of the electric field, a study from 2021” it is not clear whether it refers to the authors' previous work or some other work.”

Answer: Thank you for your question. According to the cited reference [43], the study belongs to Lee, S and his colleagues.

Comment (7): “What is the novelty of the work compared to previous similar studies?”

Answer: Thank you for your question. The novelty of this work is to study the influence of the magnetic and electric fields, and of LASER exposure directly on the EOs, in terms of composition and bioactivities. As presented in the introduction, the influence of these treatments was investigated before on seeds germination and plants development. According to our knowledge, there is only one similar study [104], that reported the effects of the exposure of two essential oils of cedarwood and tea tree to a magnetic field. 

Comment (8): “Alpha and beta with symbols”

Answer: Thank you for your suggestions. The modifications were applied throughout the manuscript.

Comment (9): “The real statistical differences between such close numbers are not clear. For example for 1 compound in cinnamon bark (Cinnamon zeylanicum) essential oils, 67.39% 67.56% 67.81% 67.58% how different are them?”

Answer: Thank you for your question. According to the chromatographs and the reported data, the differences in percentages is not significant to support the changes in the antimicrobial effects.

Comment (10): “Antimicrobial activity: which and how many works have previously investigated the activity of these essential oils on these bacteria?”

Answer: Thank you for your question. The antimicrobial activity of these oils has been highly investigated in the literature and to mention all the results would have meant to complicate the discussion part too much. For each oil, references were given regarding these effects.

Comment (11): “Table 4 is not completely visible.”

Answer: Thank you for your suggestion, we have repositioned the table.

Comment (12): “too long and difficult to follow”

Answer: Thank you for your feedback. The length of the discussion part is a consequence of the fact that EOs are complex mixtures of substances, that we’ve used three oils and three treatments, and that we’ve followed more parameters for this study. We have presented the data according to the structure introduced in the Results section.

Comment (13): “Materials and methods: “For this study were used EOs of cinnamon bark (Cinnamomum zeylanicum), patchouli (Pogostemon cablin) and geranium (Pelagornium graveleons) which were acquired from the same international company.” Which company? More details are needed.”

Answer: Thank you for your question. We have used three EOs commercially available and we consider that for the purpose of this study the name of the company of these oils is not relevant. The cinnamon bark EO is native from Sri Lanka and is extracted from true cinnamon (Ceylon cinnamon - Cinnamomum zeylanicum Blume), patchouli EO comes from the plants grown in Indonesia (Pogostemon cablin (Blanco) Benth.) and geranium EO (Pelargonium graveolens L’Hér) is obtained from Africa and Madagascar. We have used our own investigations to present the results and to draw the conclusions.

Comment (14): “Antibacterial Activity is not correct if we talk about Candida albicans.”

Answer: Thank you for your observation. We have corrected this information where it was applicable.

Sincerely,

Camelia Scheau

Round 2

Reviewer 2 Report

Comments and Suggestions for Authors

The authors have made a lot of modifications, and I suggest accept in present form.

Author Response

Comment: The authors have made a lot of modifications, and I suggest accept in present form.

Answer: The authors would like to express their sincere gratitude for your appreciation. 

Reviewer 4 Report

Comments and Suggestions for Authors

None

Author Response

The authors would like to express their sincere gratitude for your appreciation. 

Reviewer 5 Report

Comments and Suggestions for Authors

The manuscript has been significatly improved.

Author Response

Comment 1: The manuscript has been significantly improved.

Answer: The authors would like to express their sincere gratitude for your appreciation.